# Predictors of Psychological Distress and Mental Health Resource Utilization among Employees in Malaysia

**DOI:** 10.3390/ijerph18010314

**Published:** 2021-01-04

**Authors:** Caryn Mei Hsien Chan, Siew Li Ng, Serena In, Lei Hum Wee, Ching Sin Siau

**Affiliations:** 1Centre for Community Health Studies (ReaCH), Faculty of Health Sciences, University Kebangsaan Malaysia, Kuala Lumpur 50300, Malaysia; caryn@ukm.edu.my (C.M.H.C.); chingsin.siau@gmail.com (C.S.S.); 2Department of Psychology, HELP University, Bukit Damansara, Kuala Lumpur 50490, Malaysia; ng.sl@help.edu.my; 3Department of Psychology, International Medical University, Bukit Jalil, Kuala Lumpur 57000, Malaysia; serenain@imu.edu.my

**Keywords:** psychological distress, workplace health, workplace mental health, employee resource utilization

## Abstract

We sought to examine predictors of psychological distress among employees as well as the level of awareness and usage of available mental health resources by employees through their own organizations. The Malaysian Healthiest Workplace survey cross-sectional dataset was used to explore the association between psychological distress, a range of health conditions, as well as mental health resource awareness and usage in a sample of 11,356 working Malaysian adults. A multivariate logistic regression was conducted to determine predictors of high psychological distress. Comorbid illnesses that were associated with psychological distress were mental illness (OR 6.7, 95% CI 4.39–10.14, *p* = 0.001), heart conditions (OR 2.17, 95% CI 1.18–3.99, *p =* 0.012), migraines (OR 1.59, 95% CI 1.33–1.90, *p =* 0.001), bronchial asthma (OR 1.43, 95% CI 1.11–1.85, *p* = 0.006), and hypertension (OR 1.42, 95% CI 1.07–1.88, *p* = 0.016) compared to individuals with no comorbid conditions. A total of 14 out of 17 comorbid medical illnesses were associated with elevated levels of psychological distress among employees. Awareness and usage of support services and resources for mental health were associated with lower psychological distress. These findings extend the literature by providing further evidence on the link between chronic illness, occupational type, as well as awareness and use of mental health resources by psychological distress status.

## 1. Introduction

Mental health issues are among the costliest burdens faced by both organizations and governments in the developed world [1,2]. Psychological distress, which is characterized by mental and physical symptoms associated with a state of emotional suffering [3,4], is a growing public health issue in Malaysia, with analogous social and economic impact and consequences [5].

Psychological distress is often associated with poorer physical health and higher healthcare utilization, with negative consequences to both employees and employers such as decreased work engagement, increased sickness leave, as well as higher absenteeism and presenteeism [6,7]. Recognizing the need to reduce psychological distress, many organizations are implementing mental health services for their employees to facilitate better health and have made a plethora of mental health support services and resources available in response [8]. In spite of the various options available, however, there is evidence to indicate that these resources may be underutilized by the very individuals who need them most [1]. Rather than merely focusing on the provision of more services and resources, there is a need to examine which services are most acceptable, useful, and utilized in our Malaysian setting.

Much of the published literature in this area has focused on high-resource countries [9,10,11]. Thus, there are major gaps in what is known about this situation in low- to middle-resource nations, where mental health literacy and acceptance of mental health are much lower [12,13,14,15].

It is therefore important to examine mental wellbeing in the Malaysian context specifically due to its unique cultural considerations that need to be taken into account in this sample [16]. The mental healthcare system in Malaysia is still very much in its infancy, e.g., there is a lack of mental health professionals to service its growing population (e.g., a ratio of about 5 psychiatrists per 100,000 people in the capital), disproportionate distribution of available services in the country [17], and a lack of insurance coverage for mental health concerns [18]. These are coupled with the reluctance to seek help or accept treatment for mental issues due to the associated stigma, making accessibility to treatment a key challenge in this country [16].

Mental health burden in this country is rising; according to Malaysia’s 2019 National Health and Morbidity Survey, we are witnessing a trend of higher psychological distress among even children between the ages of 5 to 15 [19]. A survey carried out by the Ministry of Health showed that hospital admission for psychiatric issues has been decreasing over the years, yet the number of total outpatient psychiatric cases has overall increased [20]. However, information on utilization of mental health services in the community, outside of the hospital system, is scarce and lacking.

In terms of occupation, a study based on 728 adults from three different states in Malaysia indicated that professional-type working adults were less depressed while business owners were more depressed [21]. Another study based on the 2011 National Health and Morbidity Survey reported that Malaysian adults with lower self-rated health were more likely to be depressed and anxious [22]. Subsequent studies among Malaysian working adults who were schoolteachers and nurses further found that those reporting musculoskeletal pain had poor psychosocial work factors and higher psychological distress, such as severe anxiety and depression [23,24]. Low resource and facilities in the workplace, lower job category, and higher workplace responsibility also predicted depression and anxiety among university lecturers [25]. Overall, the literature on Malaysian working adults’ mental health is sparse or is focused on select occupations.

There is therefore a need for large-scale studies which would afford better representation and increase understanding into the nuances of health-seeking behaviors in this region when provided as an employee benefit. Also lacking is information on employee awareness of available mental health resources and to what extent they are utilized [10,26].

This formed the underlying basis and the key objective of this study, which was to determine predictors of psychological distress among employees in Malaysia. We also sought to examine the level of awareness and usage of available mental health resources by employees through their own organizations.

## 2. Materials and Methods

### 2.1. Survey Design and Participants

This was a cross-sectional questionnaire study based on data from the Malaysia’s Healthiest Workplace by AIA Vitality survey 2018, an annual online workplace survey among working adults in Malaysia. The aim of this survey was to provide employers with evidence-based data on their employees’ health and wellbeing to support employers to strategize and provide intervention improving the health goals and to improve company productivity [27].

This study is modelled after Britain’s Healthiest Workplace [28] in partnership with RAND Europe. Ethics approval was obtained from the Universiti Kebangsaan Malaysia Research Ethics Committee (NN-2019-152). A total of 117 corporate companies in Malaysia participated, with the study methodology described in detail elsewhere [6,7].

This study analyzed data obtained from the Employee Survey Questionnaire. The questionnaire covers multiple dimensions of self-reported health and wellbeing relevant to the workplace. The following are the variables of interest used in the analyses: sociodemographic information on age, gender, marital status, education, occupation, and income were tabulated.

### 2.2. Psychological Distress

The Kessler Psychological Distress Scale (K10) [29] is based on 10 items that measure the frequency of nonspecific psychological distress symptoms over the past 30 days. Respondents were asked, “During the past month, about how often did you feel: (1) tired out for no good reason; (2) nervous; (3) so nervous that nothing could calm you down; (4) hopeless; (5) restless or fidgety; (6) so restless you could not sit still; (7) sad or depressed; (8) so depressed that nothing could cheer you up; (9) everything was an effort; (10) worthless.” Items were rated on a five-point ordinal scale: all of the time (score 4), most of the time (score 3), some of the time (score 2), a little of the time (score 1), and none of the time (score 0). The total K10 score was calculated by summing all 10 items. K10 scores could range from 0 to 40, with higher scores indicating higher levels of psychological distress. A cutoff of ≤19 was applied, with participants who scored 19 and below categorized as the non-psychologically distressed group while those who scored 20 and above classified as the psychologically distressed group. The internal consistency of the Malaysian version of the K10 has been found to be acceptable [5].

### 2.3. Comorbid Illnesses

Dichotomous responses were solicited for comorbid illnesses with the question, “In the past 12 months, have you been told by your doctor that you have…” for bronchial asthma, heart conditions, diabetes mellitus, hypertension, migraines, kidney disease, and mental illness. Mental illness in employees in this context refers to lifetime long-enduring mental illness. Participants were asked dichotomous questions on whether they were diagnosed with the following: depression, anxiety, panic disorder, bipolar disorder, schizophrenia, and posttraumatic stress.

### 2.4. Occupational Categories

There was a total of nine occupational categories which were modelled after the World Health Organization’s Health and Productivity Questionnaire [30]. The response options “don’t know” and “prefer not to answer” were added.

### 2.5. Income

Income categories were grouped on the basis of individual monthly income, self-reported by the participants to the closest RM 999 bracket. We categorized income which best approximates the widely used bottom 40% (B40), middle 40% (M40), and top 20% (T20) strata for household income classification in Malaysia [31].

### 2.6. Awareness and Utilization of Mental Health Support Services and Resources

Respondents were posited the question “Are you aware of any of the following being available to you through your organization?, and asked to check off all applicable options from a list which included, among others, “workshops on physical and mental health”; “GP advice line”; and “resilience, energy, or stress management classes”. Use of services was also probed with the same list of options.

### 2.7. Statistical Analyses

This study used descriptive analysis with frequency, percentages, and Chi square for employee characteristics as well as employee awareness and utilization of available mental health services. Predictors of psychological distress were determined using multivariate logistic regression. All *p*-values of less than 0.05 were considered significant and based on two-tailed analyses. Odds ratios (ORs) with 95% confidence intervals (CIs) were calculated. All analyses were performed using SPSS version 26 (IBM Corp., Armonk, NY, USA).

### 2.8. Ethical Considerations

The study was performed in accordance with institutional ethics approval. All participants provided consent electronically. The participants were employees and not in a position of dependence to the researchers. They were informed about the study and were assured that all data would be stored securely, analyzed on a group level, and treated confidentially, i.e., no information regarding employees, employers, or workplaces would be identifiable.

## 3. Results

A total of 11,356 employees responded to the survey. There was a slight female preponderance (n = 6613; 58.2%). Slightly more than half of all participants in this sample were below 34 years of age (51.7%). Over half were married (n = 6606; 58.2%). The majority (n = 5601; 49.3%) held bachelor’s degrees. Professionals constituted the most populous occupational category (n = 3092; 27.2%). Most reported earnings of less than RM 3999 monthly (n = 4353, 42.1%). The top three most common comorbid illnesses reported were musculoskeletal disorders (n = 1732, n = 15.3), migraines (n = 1428, 12.6%), and hypertension (n = 857, 7.5%). Refer to Table 1 for the demographic characteristics of employees.

Employee awareness and utilization of available mental health services/resources made available to them through their organization were presented in frequencies and percentages. The majority of employees responded non-affirmatively in terms of both awareness and utilization. Only the number of participants who responded in the affirmative were presented here in this study. This data was further stratified by the psychological distress vs. no psychological distress groups. Chi-square was calculated based on 2 × 2 tables between awareness/utilization (yes vs. no) and psychological distress status (distress vs. not distressed). Among the services offered, awareness of volunteering or charity work was the highest (n = 1604, 14.1%, *p* = 0.001), followed by workshops on physical and mental health (n = 1399, 12.3%, *p* = 0.001), and coaching (n = 1129, 9.9%, *p* = 0.001). In terms of usage of services, the most frequently endorsed were volunteering or charity work (n = 906, 8.0%, *p* = 0.001); followed by attending workshops on physical and mental health (n = 877, 7.7%, *p* = 0.001); and use of resilience, energy, or stress management classes (n = 595, 5.2%, *p* = 0.001). Table 2 details all frequencies and percentages for employee awareness and utilization of available mental health services.

A multivariate logistic regression was conducted to determine predictors of high psychological distress. The protective effect of age as a factor appeared to reduce with increasing age, with the results for employees by age bracket as follows: 18–24 years (OR 0.14, 95% CI 0.06–0.33, *p* = 0.001), 25–34 years (OR 0.14, 95% CI 0.06–0.33, *p* = 0.001), 35–44 years (OR 0.23, 95% CI 0.06–0.33, *p* = 0.001), and 45–54 years (OR 0.36, 95% CI 0.15–0.85, *p* = 0.019) compared to those 55 years and above. Being single or unmarried predicted psychological distress (1.55 times more likely to be psychologically distressed compared to married counterparts; OR 1.55, 95% CI 1.33–1.82, *p* = 0.001).

Clerical and administrative support employees were more likely to be psychologically distressed (OR 1.64, 95% CI 1.22–2.22, *p* = 0.001), as were professionals (OR 1.40, 95% CI 1.04–1.88, *p* = 0.028) and the combined occupational categories of precision production and crafts workers, chemical/production operators, and laborers (OR 1.74, 95% CI 1.18–2.56, *p* = 0.005) compared to employees from undefined occupational categories.

Income in this sample protected against psychological distress, with higher income having less of a protective effect (less than RM 1000–3999, OR 0.56, 95% CI 0.42–0.75, *p* = 0.001; RM 4000–RM 7999, OR 0.63, 95% CI 0.49–0.81, *p* = 0.001).

Comorbid illness that were associated with psychological distress were mental illness (OR 6.7, 95% CI 4.39–10.14, *p* = 0.001), heart conditions (OR 2.17, 95% CI 1.18–3.99, *p* = 0.012), migraines (OR 1.59, 95% CI 1.33–1.90, *p* = 0.001), bronchial asthma (OR 1.43, 95% CI 1.11–1.85, *p* = 0.006), and hypertension (OR 1.42, 95% CI 1.07–1.88, *p* = 0.016). Individuals with other comorbid illnesses (OR 1.85, 95% CI 1.25–2.72, *p* = 0.002) and musculoskeletal disorders (OR 1.29, 95% CI 1.05–1.59, *p* = 0016) were also more likely to be psychologically distressed compared to those without any comorbid illness.

In terms of awareness of mental health resource available to employees through their organizations, the following appeared to serve as a protective factor for psychological distress: workshops on physical and mental health (OR 0.53, 95% CI 0.33–0.86, *p* = 0.010), massage or relaxation classes (OR 0.42, 95% CI 0.19–0.92, *p* = 0.030), employee assistance (OR 0.45, 95% CI 0.26–0.90, *p* = 0.007), and mental health and wellbeing information (OR 0.36, 95% CI 0.17–0.76, *p* = 0.007). Awareness of Cognitive Behavior Therapy (CBT) or other types of psychological therapy was associated with higher psychological distress (OR 5.08, 95% CI 2.20–11.73, *p* = 0.001).

In terms of actual utilization of services available, the following appeared to serve as a protective factor: use of massage or relaxation classes (OR 0.34, 95% CI 1.33–8.56, *p* = 0.011) and use of employee assistance (OR 0.30, 95% CI 1.53–6.82, *p* = 0.002). Table 3 lists all predictors of psychological distress among employees entered into the regression model.

## 4. Discussion

A principal finding of this large-scale study was the low utilization rates of available mental health resources, affording insight into the nuances of health-seeking behaviors among employees in Malaysia. Our findings are novel in that they highlight the poor level of employee awareness when it comes to available mental health resources. Even when individuals were aware of the availability of such services, employees may still be reluctant to actively seek help. Awareness of cognitive behavior therapy or other types of psychological therapy was associated with being over five times more likely to report higher psychological distress. Findings from this study also underscored how underutilized these services are even among a relatively urban, educated sample of employed adults in this country. What then of unemployed individuals who are even less likely to afford or have access to mental health services and support? It is easy to speculate that rates of awareness and/or utilization of such services are likely to be even more dismal among the socioeconomically disadvantaged, less educated, and individuals who lack accessibility or availability. One key contributing factor to this is the poor mental health literacy in the Malaysian context [16], where individuals tend to equate use of counselling and/or psychological therapy with severe mental illness.

We found that comorbid medical illnesses were associated with elevated levels of psychological distress among employees. Psychological distress was strongly associated with 14 out of 17 health conditions and multiple other risk factors investigated in this study. Mental illness in employees was associated with being 6.7 times more likely to be psychologically distressed than those without comorbid medical illness. The other physical illness associated with higher likelihood of reporting psychological distress were heart conditions (up to 2.2 times), migraines (up to 1.6 times), bronchial asthma (up to 1.4 times), hypertension (up to 1.4 times), and musculoskeletal disorders (up to 1.3 times). Employees who reported kidney disease, cancer, epilepsy, cerebral palsy, spina bifida, cystic fibrosis, muscular dystrophy, multiple sclerosis, and paralysis of any kind, which were classed under a combined category of other comorbid illnesses, were 1.9 times more likely to experience psychologically distress. This finding extends prior knowledge on the link between physical and mental health [32] and highlights the need for further provisions required to help employees with comorbid mental and/or physical illness receiving integrated care as well as for focus on primary prevention for healthy employees. This is particularly important as physical and mental health are usually treated separately despite the known existing link between the two conditions.

An expected but nonetheless still intriguing finding was that employees only appeared to be aware of a limited range of support services and resources for mental health. In terms of awareness of mental health resource available to employees through their organizations, awareness of several resources in particular appeared to serve as a protective factor against psychological distress. Respondents who had awareness of mental health and wellbeing information were 64 percent less likely to report psychological distress, followed by massage or relaxation classes (58 percent less likely), employee assistance (55 percent less likely), and workshops on physical and mental health (47 percent less likely). Awareness of cognitive behavior therapy or other types of psychological therapy was associated with at least five times higher psychological distress, although the latter is likely due to the fact that employees who were aware of cognitive behavior therapy were plausibly more likely to have undergone it than not. Underutilization of services is not uncommon and has been documented both in Malaysia [14] and elsewhere [33]. Current findings also showed that a majority of the employees were not aware of resources available to them, which likely affected the utilization rate of such resources. It is also possible, given the cultural context of this study, that employees may be reluctant to take up psychological therapy due to the stigma still associated with mental illness and/or usage of services targeted at alleviating it [14,16].

Even fewer numbers endorsed utilization of the available services. In terms of actual utilization of services available, only the following appeared to serve as a protective risk factor: use of employee assistance (70 percent less likely) and use of massage or relaxation classes (66 percent less likely). Only a limited range of support services and resources for mental health were endorsed by employees for awareness and usage. In terms of usage, this finding was striking because only 2 out of 18 mental health support services and resources were found to be associated with lower psychological distress among employees. Among the services offered, volunteering and charity work were the most well-known and used mental health service among employees; 14.1% of employees reported being aware of such a service, whereas 8% reportedly used it. We believe that this finding was not unexpected given the altruistic, interdependent, and collectivistic characteristics that are highly valued in this culture.

Because employees need resources to address psychological distress, we need to better understand barriers to help seeking and service use. Many psychologically distressed employees may not be aware of services other than those offered by clinical or community health settings (mainstream clinical mental health) and thus may not benefit from using these services to their full potential [10]. In addition, employees may also fear potential repercussions (e.g., breach of confidentiality) of using such services as some employers may track usage of services. There is a need to focus on improving the mental health of employees including leader support on mental health literacy and reducing stigma [2,34] rather than expanding variations of available resources. Leader support has been found to be essential in the success of work health promotion programs [2,34,35] and may also serve to promote employee psychological wellbeing and to overcome stigma and discrimination [13]. However, it is unclear whether this focus is appropriate or whether other targets should be given greater consideration, such as targeting interventions that are more acceptable to employees in this setting.

The finding that precision production and crafts workers, chemical/production operators, and laborers were 1.7 times more likely to be psychologically distressed compared to employees from undefined occupational categories is consistent with past studies which reported that workers from the blue-collar sector are at higher risk for being suicidal compared to the general working-age group [36]. Employees from clerical and administrative support occupations were also 1.6 times more likely to be psychologically distressed, as were professionals, who were 1.4 times more likely compared to employees from undefined occupational categories. These findings suggest that the workplace environment may be a facilitator of employees’ mental health and confirm previous findings [37] of the association between occupational factors and psychological distress.

Several other factors examined in this study contributed independently to explaining psychological distress. In terms of marital status, employees who were single were 1.6 times more likely to be psychologically distressed compared to their married counterparts, which is consistent with other studies, which indicates that the absence of social support, particularly in those who are living alone, may play a role in poorer mental health [38].

Employees with lower income in this sample appeared to be less likely to report psychological distress, with employees in the ≤RM 1000–3999 and RM 4000–RM 7999 income brackets being up to 44 percent and 37 percent less likely to report psychological distress compared to employees from the ≥RM 8000 income bracket. This is interestingly at odds with a study from this region which indicated that lower income is tied to greater psychological distress [39].

Younger age was associated with lower psychological distress in this sample. Employees who were in the 18–24 and 25–34 age bracket both shared an 86 percent lower likelihood of being psychologically distressed, while those from the 35–44 years and 45–54 years age brackets were 77 percent and 64 percent less likely to be psychologically distressed, respectively, compared to those 55 years and above. This is inconsistent with the general trend of findings that suggest that psychological distress decreases with age [39,40]. This may be explained by the probability that older adults are more likely to have gained greater resilience and coping skills in managing both work and life challenges with experience and maturity, lowering the risk of psychological distress. Thus, the effects of life and work stressors become less salient with increasing age. However, a positive curvilinear relationship between age and psychological distress was also found, where psychological distress rises again after its lowest point at a certain age [41]. Older adults are likely to have more comorbid illnesses, and results in the current study showed an increased risk of psychological distress among those with comorbid illnesses.

No gender differences were evident from our study, which is at odds with Halonen et al. [42], who found gender-specific mental health profiles for large occupational groups where female employees reported higher stress compared to males. Distinctive mental health profiles have also been previously linked to occupational status [37]. Studies in Malaysia have documented high stress levels among employees, especially educators [43,44] and health care workers [45,46,47,48]. This study extends what is known with regard to the mental wellbeing of Malaysian employees from other occupational sectors.

Health care professionals may lack knowledge of circumstances in workplaces and may therefore be uncertain about the specific demands and expectations for different work situations. This is especially problematic when dealing with nonspecific symptoms such as stress and anxiety [2]. Thus, a better understanding of which occupational sectors are at higher risk can help mental health professionals better understand specific occupational risk factors for psychological distress as well as tailor appropriate interventions.

Avenues for future studies should include examining reasons for psychological distress, including organizational factors (poor working conditions, deadlines, workload, job insecurity, relationship with colleagues [49], workplace environment [50], roles in organization and career development [51], and family or personal problems [52]). It is important to examine the facilitators and barriers to engaging with digital mental health interventions in the workplace [26], which was beyond the scope of this study.

As this is a cross-sectional study, recall bias may have occurred. Respondents may have been understandably reluctant to disclose their mental health status, hence the possibility that rates of psychological distress may have been underreported due to previously discussed fear of stigma and discrimination. Precautions were taken to ensure that participants identifiers were guarded, and assurance was given for all handling of data with strict confidentiality. The variation in occupational types sampled does not allow us to extrapolate. This is not surprising but reflects an inherent challenge when studying a psychological work environment. The large dataset and sample size representing the Malaysian workforce, however, increases the generalizability of this study.

## 5. Conclusions

This large-scale study allows better representation and increases insight into the awareness of mental health services and health-seeking behaviors among employees in Malaysia. Our study findings additionally highlight predictors of psychological distress in workplace health studies. Taken together, it pinpoints the need for a greater focus on mental wellness among our working population in terms of preventive measures by aiding prediction of employees most at risk of psychological distress through predictors identified in this study. Our data on service utilization can also be used in the future to help inform which interventions are most acceptable to employees in this setting if we want to improve their mental wellbeing.

## Figures and Tables

**Table 1 ijerph-18-00314-t001:** Demographic characteristics of employees (N = 11,356).

Characteristics	Psychologically Distressed (n, %)	No Psychological Distress (n, %)	Total	*p*-Value
Age categories				0.001
18–24 years	140 (12.5)	690 (6.7)	830 (7.3)	
25–34 years	667 (59.5)	4380 (42.8)	5047 (44.4)	
35–44 years	233 (20.8)	3165 (30.9)	3398 (29.9)	
45–54 years	72 (6.4)	1573 (15.4)	1645 (14.5)	
55 years and above	9 (0.8)	427 (4.2)	436 (3.8)	
Gender				0.557
Female	662 (59.1)	5951 (58.1)	6613 (58.2)	
Male	459 (40.9)	4284 (41.9)	4743 (41.8)	
Marital status				0.001
Single	616 (55.0)	3614 (35.3)	4230 (37.2)	
Married	439 (39.2)	6167 (60.3)	6606 (58.2)	
Other—cohabitating, separated, divorced, widowed	66 (5.9)	454 (4.4)	520 (4.6)	
Education				0.001
No formal education or lower than secondary school completion	139 (12.4)	1459 (14.3)	1598 (14.1)	
Post-secondary school completion	242 (21.6)	2303 (22.5)	2544 (22.4)	
Undergraduate degree	620 (55.3)	4981 (48.7)	5601 (49.3)	
Postgraduate degree	120 (10.7)	1493 (14.6)	1613 (14.2)	
Occupational categories				0.001
Executive, administrator, or senior manager	27 (2.4)	237 (2.3)	264 (2.3)	
Professional	317 (28.3)	2775 (27.1)	3092 (27.2)	
Clerical and administrative support	216 (19.3)	2313 (22.6)	2529 (22.3)	
Sales	205 (18.3)	2505 (24.5)	2710 (23.9)	
Technical support	155 (13.8)	1031 (10.1)	1186 (10.4)	
Service occupation	47 (4.2)	375 (3.7)	422 (3.7)	
Combined all other	58 (5.2)	531 (5.2)	589 (5.2)	
Do not know or prefer not to answer	96 (8.6)	468 (4.6)	564 (5.0)	
Income				0.001
Less than RM 1000 to RM 3999	578 (56.4)	3775 (40.5)	4353 (42.1)	
RM 4000 to RM 7999	343 (33.5)	3207 (34.4)	3550 (34.3)	
RM 8000 and above	104 (10.1)	2328 (25.0)	2432 (23.5)	
Comorbid illness				
Bronchial asthma	91 (8.1)	503 (4.9)	594 (5.2)	0.001
Heart conditions	16 (1.4)	92 (0.9)	108 (1.0)	0.084
Diabetes mellitus	19 (1.7)	341 (3.3)	360 (3.2)	0.003
Hypertension	75 (6.7)	782 (7.6)	857 (7.5)	0.253
Migraines	218 (19.4)	1210 (11.8)	1428 (12.6)	0.001
Arthritis	29 (2.6)	265 (2.6)	294 (2.6)	0.997
Mental illness	55 (4.9)	64 (0.6)	119 (1.0)	0.001
Other comorbidities combined	39 (3.5)	216 (2.1)	255 (2.2)	0.003
Musculoskeletal disorders	140 (12.5)	1592 (15.6)	1732 (15.3)	0.007

Mental illness in this context refers to long-enduring mental illness (depression, anxiety, panic disorder, bipolar disorder, schizophrenia, and posttraumatic stress). Other comorbidities combined include kidney disease (n = 56), cancer (n = 27), epilepsy (n = 17), cerebral palsy (n = 3), spina bifida (n = 20), cystic fibrosis (n = 56), muscular dystrophy (n = 71), multiple sclerosis (n = 8), and paralysis of any kind (n = 7).

**Table 2 ijerph-18-00314-t002:** Employee awareness and utilization of available mental health services (N = 11,356).

Variable	Psychologically Distressed (n, %)	No Psychological Distress (n, %)	N (%)	*p*-Value
**Awareness of services**				
Workshops on physical and mental health	55 (4.9)	1344 (13.1)	1399 (12.3)	0.001
GP advice line	38 (3.4)	544 (5.3)	582 (5.1)	0.006
Training on common mental health conditions (such as depression, anxiety disorders etc.)	74 (6.6)	926 (9.0)	1000 (8.8)	0.006
Resilience, energy, or stress management classes	52 (4.6)	988 (9.7)	1040 (9.2)	0.001
Mindfulness classes	42 (3.7)	662 (6.5)	704 (6.2)	0.001
Massage or relaxation classes	30 (2.7)	441 (4.3)	471 (4.1)	0.009
Counselling or psychotherapy services	44 (3.9)	877 (8.6)	921 (8.1)	0.001
Employee assistance	41 (3.7)	785 (7.7)	826 (7.3)	0.001
Workload or time management training	53 (4.7)	691 (6.8)	744 (6.6)	0.009
Volunteering or charity work	93 (8.3)	1511 (14.8)	1604 (14.1)	0.001
Coaching	60 (5.4)	1069 (10.4)	1129 (9.9)	0.001
Mental health support onsite/telephone/mobile app/online	17 (1.5)	346 (3.4)	363 (3.2)	0.001
CBT or other types of psychological therapy	17 (1.5)	229 (2.2)	246 (2.2)	0.115
Mental health and wellbeing information	33 (2.9)	780 (7.6)	813 (7.2)	0.001
Financial wellbeing courses	38 (3.4)	861 (8.4)	899 (7.9)	0.001
Wellbeing app targeting a broad range of physical health, mental health and lifestyle issues	76 (6.8)	807 (7.9)	883 (7.8)	0.190
Wellbeing app targeting specific health issues, such as weight, exercise, or mental health	57 (5.1)	664 (6.5)	721 (6.3)	0.067
Online coaching	30 (2.7)	277 (2.7)	307 (2.7)	0.953
**Use of services**				
Use of workshops on physical and mental health issues in past 12 months	32 (2.9)	845 (8.3)	877 (7.7)	0.001
Use of GP advice line	17 (1.5)	286 (2.8)	303 (2.7)	0.012
Use of training on common mental health conditions (such as depression, anxiety disorders etc.)	27 (2.4)	421 (4.1)	448 (3.9)	0.005
Use of resilience, energy, or stress management classes	23 (2.1)	572 (5.6)	595 (5.2)	0.001
Use of mindfulness classes	24 (2.1)	324 (3.2)	348 (3.1)	0.059
Use of massage or relaxation classes	20 (1.8)	197 (1.9)	217 (1.9)	0.744
Use of counselling or psychotherapy services	16 (1.4)	295 (2.9)	311 (2.7)	0.005
Use of employee assistance	25 (2.2)	249 (2.4)	274 (2.4)	0.675
Use of workload or time management training	32 (2.9)	313 (3.1)	345 (3.0)	0.706
Use of volunteering or charity work	48 (4.3)	858 (8.4)	906 (8.0)	0.001
Use of coaching	29 (2.6)	478 (4.7)	507 (4.5)	0.001
Use of other mental health support onsite/ telephone/mobile app/online	9 (0.8)	120 (1.2)	129 (1.1)	0.268
Use of CBT or other types of psychological therapy	5 (0.4)	81 (0.8)	86 (0.8)	0.205
Use of mental health and wellbeing information	20 (1.8)	441 (4.3)	461 (4.1)	0.001
Use of financial wellbeing courses	20 (1.8)	485 (4.7)	505 (4.4)	0.001
Use of wellbeing app targeting a BROAD range of physical health, mental health, and lifestyle issues	51 (4.5)	544 (5.3)	595 (5.2)	0.275
Use of wellbeing app targeting *specific* health issues, such as weight, exercise, or mental health	39 (3.5)	437 (4.3)	476 (4.2)	0.210
Use of online coaching	16 (1.4)	137 (1.3)	153 (1.3)	0.807

Based on the question “Are you aware of any of the following being available to you through your organization?” Note that, for both psych distressed and non-psychologically distressed employees, only “yes” responses were recorded. Percentages are based on the “yes” responses. The *p*-value is based on Chi-square differences between yes and no responders. Column percentages were used for psychological distress and non-psychologically distressed groups.

**Table 3 ijerph-18-00314-t003:** Predictors for K10 psychological distress using a score of 19 as the cutoff (N = 11,356).

	B	S.E.	Wald	Sig.	Exp(B)	95% C.I.
Age categories			50.87	0.001		
18–24 years	−1.99	0.45	19.86	0.001	0.14	0.06–0.33
25–34 years	−1.96	0.43	20.49	0.001	0.14	0.06–0.33
35–44 years	−1.48	0.45	11.99	0.001	0.23	0.10–0.53
45–54 years	−1.03	0.44	5.48	0.019	0.36	0.15–0.85
55 years and above (Ref.)	–	–	–	–	–	–
Gender						
Female	−0.13	0.07	3.09	0.079	0.88	0.76–1.02
Male (Ref.)	–	–	–	–	–	–
Marital status			35.03	0.000		
Single	0.44	0.08	30.07	0.001	1.56	1.33–1.82
Married	−0.12	0.17	0.46	0.497	0.89	0.63–1.25
Other—cohabitating, separated, divorced, widowed (Ref.)	–	–	–	–	–	–
Education			3.83	0.280		
No formal education or lower than secondary school completion	−0.27	0.16	2.97	0.085	0.76	0.56–1.04
Post-secondary school completion	−0.18	0.14	1.78	0.182	0.83	0.64–1.09
Undergraduate degree	−0.21	0.12	3.21	0.073	0.81	0.65–1.02
Postgraduate degree (Ref.)	–	–	–	–	–	–
Occupational categories			17.07	0.017		
Executive, administrator, or senior manager	0.29	0.25	1.34	0.247	1.34	0.82–2.19
Professional	0.33	0.15	4.86	0.028	1.40	1.04–1.88
Clerical and administrative support	0.50	0.15	10.39	0.001	1.64	1.22–2.22
Sales	0.19	0.16	1.32	0.251	1.21	0.88–1.66
Technical support	0.20	0.16	1.47	0.226	1.22	0.89–1.68
Service occupation	0.39	0.21	3.45	0.063	1.48	0.98–2.24
Combined all other	0.55	0.20	7.91	0.005	1.74	1.18–2.56
Do not know or prefer not to answer	–	–	–	–	–	–
Income			16.13	0.001		
Less than RM 1000 to RM 3999	−0.58	0.15	15.81	0.001	0.56	0.42–0.75
RM 4000 to RM 7999	−0.46	0.13	12.56	0.001	0.63	0.49–0.81
RM 8000 and above (Ref.)	–	–	–	–	–	–
Comorbid illness						
Bronchial asthma	0.36	0.13	7.52	0.006	1.43	1.11–1.85
Heart conditions	0.78	0.31	6.26	0.012	2.17	1.18–3.99
Diabetes mellitus	−0.27	0.27	1.04	0.309	0.76	0.45–1.29
Hypertension	0.35	0.14	5.82	0.016	1.42	1.07–1.88
Migraines	0.47	0.09	26.07	0.001	1.59	1.33–1.90
Arthritis	0.21	0.22	0.86	0.353	1.23	0.79–1.91
Mental illness	1.90	0.21	79.01	0.001	6.68	4.39–10.15
Other comorbidities combined	0.61	0.20	9.61	0.002	1.85	1.25–2.72
Musculoskeletal disorders	0.26	0.11	5.77	0.016	1.29	1.05–1.59
No comorbid illness (Ref.)						
Level of awareness of the service types available through the employee’s organization						
Workshops on physical and mental health	−0.64	0.25	6.70	0.010	0.53	0.33–0.86
GP advice line	0.29	0.27	1.15	0.283	1.34	0.79–2.26
Training on common mental health conditions (such as depression, anxiety disorders etc.)	0.25	0.20	1.66	0.197	1.29	0.88–1.89
Resilience, energy or stress management classes	−0.06	0.24	0.07	0.796	0.94	0.58–1.51
Mindfulness classes	−0.06	0.29	0.04	0.833	0.94	0.53–1.66
Massage or relaxation classes	−0.86	0.40	4.71	0.030	0.42	0.19–0.92
Counselling or psychotherapy services	−0.21	0.23	0.83	0.361	0.81	0.51–1.28
Employee assistance	−0.79	0.29	7.35	0.007	0.45	0.26–0.80
Workload or time management training	0.14	0.26	0.29	0.591	1.15	0.69–1.91
Volunteering or charity work	−0.12	0.18	0.43	0.513	0.89	0.63–1.26
Coaching	−0.11	0.22	0.26	0.613	0.90	0.58–1.38
Mental health support onsite/telephone/mobile app/online	−0.89	0.47	3.54	0.060	0.41	0.16–1.04
CBT or other types of psychological therapy	1.62	0.43	14.45	0.001	5.08	2.20–11.73
Mental health and wellbeing information	−1.02	0.38	7.29	0.007	0.36	0.17–0.76
Financial wellbeing courses	−0.27	0.29	0.84	0.361	0.77	0.43–1.36
Wellbeing app targeting a broad range of physical health, mental health, and lifestyle issues	0.16	0.28	0.34	0.558	1.18	0.68–2.04
Wellbeing app targeting specific health issues, such as weight, exercise, or mental health	0.01	0.32	0.01	0.983	1.01	0.54–1.89
Online coaching	0.62	0.36	2.99	0.083	1.85	0.92–3.72
Utilization of services						
Use of workshops on physical and mental health issues in past 12 months	−0.08	0.31	0.06	0.800	0.92	0.50–1.71
Use of GP advice line	−0.53	0.39	1.88	0.170	0.59	0.27–1.26
Use of training on common mental health conditions (such as depression, anxiety disorders etc.)	−0.27	0.30	0.81	0.370	0.77	0.43–1.37
Use of resilience, energy, or stress management classes	−0.57	0.35	2.74	0.098	0.56	0.29–1.11
Use of mindfulness classes	0.33	0.38	0.74	0.390	1.39	0.66–2.93
Use of massage or relaxation classes	1.22	0.48	6.52	0.011	3.37	1.33–8.56
Use of counselling or psychotherapy services	0.16	0.38	0.18	0.673	1.18	0.56–2.48
Use of employee assistance	1.17	0.38	9.47	0.002	3.23	1.53–6.82
Use of workload or time management training	0.24	0.34	0.49	0.483	1.27	0.65–2.46
Use of volunteering or charity work	−0.31	0.24	1.72	0.190	0.73	0.46–1.17
Use of coaching	−0.18	0.31	0.34	0.560	0.83	0.45–1.54
Use of other mental health support onsite/telephone/mobile app/ online	0.82	0.63	1.72	0.190	2.28	0.67–7.80
Use of CBT or other types of psychological therapy	−1.01	0.71	2.03	0.154	0.37	0.09–1.46
Use of mental health and wellbeing information	0.62	0.45	1.94	0.164	1.86	0.78–4.47
Use of financial wellbeing courses	−0.04	0.39	0.01	0.910	0.96	0.45–2.04
Use of wellbeing app targeting a BROAD range of physical health, mental health and lifestyle issues	−0.14	0.33	0.17	0.679	0.87	0.46–1.67
Use of wellbeing app targeting SPECIFIC health issues, such as weight, exercise, or mental health	0.06	0.38	0.03	0.871	1.06	0.51–2.22
Use of online coaching	−0.26	0.47	0.30	0.584	0.77	0.31–1.93
Constant	0.39	0.89	0.20	0.658	1.48	

Ref.: Reference group.

## Data Availability

Participants’ personal information is solely held by RAND Europe CIC and the data collection partner, Survey Sampling UK Ltd. (SSI). All of participants’ personal data is confidential and cannot be shared with third parties.

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
