# Peer review of "Predictors of Psychological Distress and Mental Health Resource Utilization among Employees in Malaysia"

_ijerph, 2021, doi:10.3390/ijerph18010314_

Round 1

Reviewer 1 Report

Thanks for letting me review your paper.  I hope these comments help to improve your paper.  Once they are addressed, I believed this paper could be published.  Thank you.

Comments:

Add aspects of Malaysia and culture especially in regards to mental health in the Introduction section of the paper.  A paragraph will do.

Delete first three sentences of your Discussion section.

Age, SES, and marital status are typical results of this study.  So, please write a paragraph on what is novel about the results of these findings.  Again, place this information within a cultural context.

Write more in Conclusion.  Be specific about what you found, how it is relevant, and how it can change things in the future.  Right now, it says nothing novel to me.

Reviewer 2 Report

The language used and organization of the article are weird. There are quite many grammatical errors in the article. Besides, extraordinarily short paragraphs are found in the introduction and conclusion sections.

A thorough literature review is lacking. The authors' claimed dearth of related literature in low-middle resource nations (lines 47-48) is highly doubtful. The impetus of the article/study is not justified.

The research drew upon the AIA Vitality survey 2018. However, the sampling strategy and methodology of that survey are not discussed in the article. There is no way for the reviewer to comment on the representativeness of the sample and generalizability of the research findings. 

More importantly, significant parts of the text are similar (or copied) from other works or sources, particularly

  • https://www.suffolkchamber.co.uk/media/46330/not-just-free-fruit-health-and-wellbeing-at-work.pdf
  • https://www.mdpi.com/2072-6643/12/1/50/review_report

Also, these sources are not cited in the paper at all.

Round 2

Reviewer 1 Report

My comments have been addressed.  Thank you.

Reviewer 2 Report

I don't have further comments on the paper.